# UK Nurses Delivering Physical Activity Advice: What Are the Challenges and Possible Solutions? A Qualitative Study

**DOI:** 10.3390/ijerph20237113

**Published:** 2023-11-27

**Authors:** Anoop Raghavan, Dane Vishnubala, Adil Iqbal, Ruth Hunter, Katherine Marino, David Eastwood, Camilla Nykjaer, Andy Pringle

**Affiliations:** 1York City Football Club, York YO32 9AF, UK; 2School of Biomedical Sciences, Faculty of Biological Sciences, University of Leeds, Leeds LS2 9JT, UK; d.vishnubala@leeds.ac.uk (D.V.); dr.david.eastwood@gmail.com (D.E.); c.nykjaer@leeds.ac.uk (C.N.); 3School of Public Health, Imperial College London, London SW7 2BX, UK; 4Calderdale and Huddersfield NHS Foundation Trust, Huddersfield HD3 3EA, UK; adil.iqbal@doctors.org.uk; 5York and Scarborough Teaching Hospitals NHS Foundation Trust, York YO31 8HE, UK; ruth.hunter6@nhs.net; 6Royal Stoke University Hospital, Stoke-on-Trent ST4 6GQ, UK; katiemarino@live.co.uk; 7Musculoskeletal Department, Locala Health and Wellbeing, Huddersfield HD1 4EW, UK; 8Clinical Exercise and Rehabilitation Research Centre, School of Sport and Exercise Science, University of Derby, Derby DE22 1GB, UK; a.pringle@derby.ac.uk

**Keywords:** physical activity, nurses, knowledge, awareness, advice

## Abstract

There are a multitude of health benefits gained from regular physical activity (PA). Currently, PA advice implementation from NHS nurses is inadequate despite their ever-increasing role in lifestyle and preventive medicine. By assessing their knowledge of current PA guidance, this study proposed to investigate the issues with regular PA advice being given and expand upon nurses’ proposed barriers and solutions. A qualitative approach using semi structured interviews was undertaken between March and August 2023 involving 13 NHS nurses. Thematic analysis was undertaken using Braun and Clarke’s six step approach. Four themes and fifteen subthemes emerged as barriers and solutions in delivering PA advice. Intrinsic barriers included a lack of nurse knowledge on the topic and PA being seen as an afterthought. Extrinsic barriers included time pressures and a lack of staff engagement. Solutions involved increasing staff awareness of guidelines through teaching, policy, encouraging staff to be active and optimising PA advice delivery through a piecemeal approach and utilising online and visual resources. This study displayed an insight into nurses’ thoughts on their consultations with patients regarding PA, and proposed several barriers and solutions. Further work is needed to improve nurses’ PA knowledge and to assess the proposed strategies to improve its delivery.

## 1. Introduction

The importance of physical activity (PA) is well understood in the realm of public health [1,2]. Recognised multisystem benefits include cardiovascular, musculoskeletal, respiratory and mental health improvements [1]. Despite this, there remain multiple challenges facing public engagement. National guidance in the United Kingdom (UK) advises a minimum of 150 min of moderate or 75 min of vigorous PA (or a combination thereof) as well as two days per week of muscle strength-based activity to reap health benefits [3]. According to the most recent Active Lives survey [4], an estimated 37% of UK adults aged 19–64 years failed to meet these standards, with 26% deemed physically inactive, completing less than 30 min of activities weekly. Not only is this cohort excluded from the established health benefits of PA, they are at risk of further independent medical comorbidities [5], most notably metabolic syndrome and mental health problems [6,7].

Both the financial and patient burden placed on the National Health Service (NHS) through direct resource funding for admissions, medications and surgical services are well established [8]. This has been impacted by the recent and unprecedented COVID-19 pandemic [9], which has disrupted established routines and potential opportunities for PA for many. The effects of wide scale gym closures, lockdown restrictions, physical deconditioning and a subsequent reluctance to engage with regular PA have all been difficulties when trying to meet national PA guidance [10]. Damaging effects not only include an increase in cardiovascular disease burden but also depression and seclusion for many [11], thus widening the gap for unmet needs.

Nurses are frontline NHS staff and increasingly work independently to manage a host of patients. A greater number of primary care practices are utilising nursing skills in roles to fulfil unmet population needs, including addressing lifestyle factors [12]. Despite being routinely involved in counselling patients with weight loss and PA advice, there remain many barriers to the distribution of routine and comprehensive PA advice, with a perceived lack of education, experience and relevance commonly stated [13]. The nature of consultations has also exacerbated this difficulty. There are greater pressures due to appointment numbers, waiting times and staff turnover, meaning less continuity of care [14]. Bridging this gap is a key focus for healthcare delivery, especially when factoring in strategies such as ‘Make Every Contact Count’ [15]. This can be summarised as aiming to deliver key PA guidance in minutes: factoring in patient baselines, limitations and PA preferences. There is a need for research into the barriers and facilitators that nurses encounter when providing PA advice. Previous studies which have explored this in a range of healthcare professions have done so using a qualitative approach [16,17]. This is due to the essential human factors and complexities surrounding PA uptake and compliance issues. Here, a greater insight is gained through qualitative methods which permit a greater candour from interviewees and enhanced depth for discussion points [18]. This iterative approach is important in providing solutions for the PA challenge described above. As such, this study seeks to establish an understanding of nurse’s awareness of PA guidelines and their views on barriers and solutions for engaging in PA consultation with their patients.

## 2. Materials and Methods

A qualitative study was conducted using semi-structured interviews and standards for reporting qualitative research [16]. The research approach was informed by other studies exploring healthcare professionals’ (HCPs) perceptions towards providing PA advice to patients [17,18]. Ethical approval was provided by the Faculty of Biological Sciences at the University of Leeds (27 July 2020/BIOSCI 19-039).

### 2.1. Participants

The inclusion criteria for this study included nurses working in the NHS, who had an up-to-date, valid license from the Nursing and Midwifery Council (NMC) and were currently working in practice at the time of interview in any clinical field. Participants were excluded if they were non-patient facing in their role. Participants were notified about the study via email, notices within NHS nursing teams and word of mouth. Those who expressed interest emailed the interviewers directly and were subsequently sent further information, including an information sheet outlining the aims of the study and what was expected of them, including the right to withdraw at any time, as well as a consent form which was signed by the participant and returned; then an interview date was arranged. Consent was again verified verbally at the start of the interview and the aims of the research were clarified. Interviews were conducted until data saturation was achieved; this was defined as the point in time when no new themes were identified from the data collection process.

### 2.2. Data Collection

Data were collected using semi-structured interviews. An open style question and answer format was used with the main advantages being the opportunity for both sides to illustrate their thoughts more clearly. Given that attitudes and ideas are a key factor in this study, a detailed set of questions were issued, with opportunities and prompts used at certain points. This conversational style helped to harbour an honest environment [19] and allowed interviewees to bring up new ideas, which is crucial for identifying real or perceived barriers. The interview guide (see Appendix A) was adapted from Vishnubala et al. (2022) [17] to make the questions specific to nurses, and included questions regarding PA knowledge and education and PA resources and interventions, as well as questions regarding the participant’s own PA in relation to the CMO PA guidance. Participants were also asked about their years of work experience, region and working field.

Interviews took place between March 2022 and August 2023 via the online meeting platform Zoom. Three separate researchers conducted the interviews. All were trained in order to attain standardisation between results via familiarisation with the interview sheet and mock interviews with DV. This provided similarity and consistency between the interview styles and helped to mitigate against interviewer bias. No financial incentives were advertised or given and there were no conflicts of interest for participation. This allowed for clear and anonymised information to be given without pressure. Interviews were automatically transcribed verbatim by Zoom, and recordings and transcripts were saved and stored securely on the University of Leeds OneDrive. Participants were anonymised by assigning them a participant number (e.g., Participant 1), with any identifiable information anonymised. This methodology was openly always shared with all participants in the study.

### 2.3. Data Analysis

Thematic data analysis was undertaken, using Braun and Clarkes 6 step approach [20]. This served to familiarise data, create codes, locate themes, revise themes, name them and, finally, report them. This provided a structured framework from which we could evaluate large sets of data. Transcripts were chronologically analysed for key features of challenges and solutions as per the aim of the study. These features were refined and coded to produce themes and subthemes. New information was reflected on and analysed via a recursive method, permitting the addition to or revision of previous data [21]. Microsoft Word transcription software was used to obtain the recording information and visually display it for analysis. Themes and subthemes were categorised, and this information was presented in tables alongside example quotes from participants. Signs of data saturation [22] were reached by the tenth participant, and there were no further themes present by the thirteenth and final participant. Microsoft Excel was used to categorise and present the information which displayed further cohort characteristics such as demographics, experience and role, as well as highlight the numbers and percentages of answers given in the interviews. A second researcher (KM) independently reviewed 5 randomly selected transcripts, and a 78% similarity in the coding was found.

## 3. Results

### 3.1. Participant Characteristics

Thirteen individuals expressed an interest in participation. As all were eligible, a total of 13 participants were included in the study. Interview length ranged between 10 min and 70 min, with an average of 34 min. Data saturation was reached at participant 13; no more emerging themes were identified beyond this sample, and no further participants were recruited.

Demographic characteristics of the 13 participants can be found in Table 1 below. The participants who did disclose their location were mainly situated in Scotland and the West Midlands. Sixty percent of the interviewees were staff nurses, and the participants were deployed across different departments ranging from cardiology to the COVID-19 assessment ward. Eighty-five percent of the interviewees had more than 6 years of experience, while 54% had received some form of PA-related training during their undergraduate studies. Almost all participants were physically active, 92% meeting the CMO PA guidelines. By contrast, 23% of participants were aware of the CMO PA guidelines; only one participant (the lead nurse on public health) was aware of the 2019 update to the PA guidelines and two respondents were familiar with the resource Moving Medicine. When presented with the CMO PA guidelines in the interview, 92% of nurses stated that their current activity met the 150 min of moderate or 75 min of vigorous intensity aerobic activity prescribed by the guidelines or a combination of the two.

### 3.2. Themes from Thematic Analysis

Following thematic analysis of the data set, four themes were identified as barriers and solutions to promoting and delivering PA guidance summarised in Table 2 below. Within the four themes, fifteen subthemes were also identified from the transcripts.

#### 3.2.1. Theme 1: Nurses Intrinsic Barriers

Over half of the nurses were confident that they could deliver the PA guidelines to patients. However, 77% of respondents professed that they were not aware of the CMO’s PA guidelines in any of its forms, nor were they acquainted with the accompanying resources such as infographics or posters. The lack of knowledge of the PA guidelines was frequently cited as a reason behind nurses lacking the confidence to deliver PA advice to patients, many expressed fears that they would give the incorrect advice. Several respondents admitted that they regularly deferred this to perceived specialists. The nurses admitted that this deference to physiotherapists, dietitians and other experts had led to “compartmentalisation” and they had come to perceive PA advice as outside their remit. Another subtheme touched upon in the transcripts was the concept of “nurse’s guilt”, referring to the decline in enthusiasm from some nurses who expressed regret after being reminded that they could have done more to support a patient (Table 3).

#### 3.2.2. Theme 2: Extrinsic Barriers to Delivering PA Advice

Multiple barriers focusing on the lack of focus given to PA were raised by the nurses (Table 4). Over half (54%) of respondents had received some form of PA-related education; however, of the nurses who responded in the affirmative, all were dissatisfied with the degree of PA education they had received during their time at university. The few respondents who had taken personal interest in the subject pursued knowledge through private study or a hobby. Throughout the transcripts, time constraints were identified as a barrier to the nurses delivering PA advice, as one interviewee succinctly stated: ‘I know if I get stuck in a conversation with a patient, I’m going to be there forever talking to them.’ Time pressure was cited as one reason for the lack of PA-related training in nurses’ CPD; one respondent observed there was a disparity in staff background in PA knowledge evidenced by some nurses having little knowledge on the matter and some actively teach colleagues and patients about several of the benefits of PA from both a personal and formal public health angle. Another nurse mentioned that a high turnover of nurses, particularly international nurses, made it difficult to enforce a uniform standard as many would move on before new information could be effectively absorbed. Furthermore, patient interest and engagement in improving their own PA levels was also a recurring challenge mentioned in the interviews. One interviewee observed that patients would only acknowledge the severity of their illness and the necessity of PA when they were situated into a PA-related program, e.g., cardiac rehab, at which point they would become more receptive to the advice.

#### 3.2.3. Theme 3: Increasing Staff Awareness of PA Guidelines

Following on from the previous theme’s lack of formal education and training, the majority of nurses were in chorus when it came to increasing staffs’ overall levels of PA-related education (Table 5). A variety of proposed solutions were provided in the interviews, whether that be integrating it into the induction training, e-learning modules, case studies, workshops or irregular afternoon courses. Several nurses asserted that learning the PA guidelines should be made mandatory or even policy. A question in the interview asked whether or not less enthusiastic HCPs should engage with PA advice delivery, the participants resoundingly agreed that they should (85%). The majority of the interviewees declared that providing PA advice was a “professional responsibility”, their “duty of care” and that all nurses should be involved. One nurse commented that not engaging in PA advice was “unfair to the patients”. Emphasising the importance of PA, one nurse asserted that “even a little bit of knowledge…opens up to potentially do further things”; another nurse proposed that less enthusiastic staff should be actively targeted for recruitment because their journeys would be inspiring. It was noted that, if they could be inspired to change, this would have a greater effect on helping a wider set of NHS staff who may be currently struggling with inactivity issues. The one respondent who disagreed (with recruiting less enthusiastic personnel), argued: “it’s very evident their heart isn’t in it...you do need to have people who’re very committed to it rather than leaving it to people who really don’t see the problem”.

#### 3.2.4. Theme 4: Optimising PA Advice Delivery

Various approaches to optimising PA advice delivery were proposed throughout the interviews (Table 6). Firstly, according to the majority of interviewees, optimal PA advice must be tailored to the individual needs of the patient. Frequent mentions of the style of communication, often supportive and encouraging, was also observed as a crucial factor to successful PA advice. Secondly a piecemeal approach, advocating patience with building up the patient’s PA level gradually, was reiterated by several interviewees in the interviews. Expanding on this, several nurses expressed that a successful approach when motivating patients to think about PA was a delicate process that requires establishing a frame of reference in the patients’ minds that is not daunting, and to provide a model of “someone who has made it to the other side” who demonstrates that the goals are achievable. For this purpose, it was recommended that model volunteers who are enthusiastic about PA should be recruited to encourage patients. The third approach recommended by the nurses was an interdisciplinary team approach encouraging closer collaboration with doctors and physiotherapists. Finally, in regard to successful PA advice delivery, to buttress the nurse’s points, accompanying resources in visual or online learning and local exercise schemes were frequently recommended in the discussions.

## 4. Discussion

The purpose of the study was to explore nurse’s awareness of PA guidelines and their views on barriers and solutions in delivering PA guidance to patients. The key findings identify issues with nurses’ baseline knowledge and effective strategies for them to implement this information into their daily practice. The barriers and solutions emerged as four main themes which were further subdivided into subthemes, thus granting a structured approach to facilitate discussion for greater appreciation of the findings [23]. Highlighted themes included barriers such as inadequate knowledge, guilt towards doing harm, compartmentalising their role and time pressures. Listed solutions included formal staff training, staff support and encouragement, interdisciplinary working and utilising resources such as government campaigns and advertising from established and newer agencies. Physical activity advice is agreed to be currently under-delivered and exploring these central reasons is key in improving both the nursing and staff experience.

### 4.1. Barriers

Inadequate knowledge toward PA guidance was commonly listed amongst the nursing cohort interviewed, which is reflected in existing studies [17,18]. Despite this study having a greater personal uptake of PA guidance (92%) compared with a recent study of UK-based doctors (*N* = 15) who reported an 80% uptake [17], there were similar levels of awareness of the guidance to those reported in similar studies of UK HCPs [17,18]. This tied in with seeing PA advice as an afterthought. Several nurses who were interviewed commented that not being aware of the PA guidance, let alone having a deep and practical knowledge, meant that factoring this into impromptu, squeezed-in consultations felt inadequate and rushed, especially in primary care. This is reflected in a 2013 study of 34 GPs, which showed that consultations with time pressures lead to fewer consults including a lifestyle discussion compared to those without time pressures [24]. Given the wealth of evidence linking regular PA and improved health outcomes, whether this should be prioritised by both undergraduate programmes as well as within routine nursing clinical duties should be reviewed. As with all forms of training, PA advice should be taught, refreshed and applied for staff to feel comfortable and competent in their role. This also tied in with the subtheme of compartmentalisation, which displayed that nurses were wary of extending beyond their remit. A 2010 integrative review showed that nurses working in general practice often sought to gain the doctors’ opinion when they perceived a gap in their knowledge base [25]. Doing no harm is and should be seen as the top priority for HCPs. It is therefore unsurprising that nurses may defer to colleagues who they deem more capable of giving PA advice.

Several participants stated that they could either not recall any lessons on PA or could recall a minimal amount of content only. Undergraduate training for nurses is vast but is lacking here [26], and through the better integration of PA concepts, physiology, health benefits and practical implementation approaches there will be a much better equipped workforce. Similar findings were presented in assessing medical school content on exercise medicine [27]. It is vital, therefore, that nurses are trained to have this information and skill as their colleagues may themselves be lacking here.

Guilt was a more subtle subtheme, but several nurses expressed the challenge of accepting new information for routine consults when this had previously been near absent. This featured as either not supplying PA advice among many consultations or giving incorrect advice. Participants in a 2019 thematic analysis of semi structured interviews from 10 UK midwives displayed a similar tendency; despite being “seen as the fountain of all knowledge” there was a gap between individual agendas from past experience, e.g., “screening or exercise” [28]. Subsequently, participants revealed challenges in reflecting on subpar past consults as this exacerbated the feeling of guilt and inadequacy now that they were competent. There is evidence that more challenging topics of conversation such as weight and activity are avoided for fear of damaging this trust [29]. It is imperative, therefore, that nurses feel supported and empowered in this crucial role.

### 4.2. Solutions

Fortunately, there were a variety of solutions raised in the study. Formal knowledge can be improved via structured teaching. Empowering staff through formal undergraduate training has been discussed, and expanding on this, postgraduate training on knowledge and skills, workplace incentives and assessments are seen as potential aids. High nurse turnover, shift work and understaffing mean that work can feel more ad-hoc rather than planned. Creating formal structures and rewards are necessary to aid this [30]. Examples include e-learning modules, formal, exercise proformas and Quality and Outcomes Framework (QOF) style points, especially considering the peak prevalence of sedentary lifestyle-associated conditions such as diabetes, hypertension and depression [31]. This can help to equalise PA to the same status as other lifestyle factors like diet, smoking, alcohol and obesity, which are often prioritised and have been embedded more successfully within the psyche of HCPs as a whole [32]. Utilising policy was another subtheme, and measures such as PA advice prior to discharge planning was raised. Creating a formalised framework in the clinical environment is thought to increase PA advice uptake, and automatic submissions for highlighted patient groups can also improve this.

The vast contingent of participants met the guidelines of PA (92%). This provided further insight into staff perceptions regarding its merits, namely physical health, mental health and structural and time management [33]. Staff encouragement and support emerged as a consistent subtheme and reflected that they would feel more enabled to offer support to patients when they themselves met these targets. A 2018 nationwide cohort study in Norway showed that the more physically active the doctors were, the more often they counselled patients regarding PA [34]. This would allow for a less dictatorial environment, which is, unfortunately, common in medical interactions [35]. There have been similar findings within cohorts of physically active doctors being more likely to offer PA advice routinely, which was found to be particularly useful when targeted towards patient comorbidities [36]. This should serve to support nurses who can feel afraid to discuss PA strategies as although the topic often leads patient interest, a fear of perceived stigma prevents regular consultation [37]. There is a greater conversation regarding the ethical implications of offering advice, which is not personally taken up by medical professionals; however, this conflict was not significantly challenged by this study. Supporting staff can be achieved via showing them the value of PA from a ground level through staff awards for those greatly achieving PA targets, challenges such as “cycle to work” programmes [38] and profile interviews with successful team members explaining the successful daily strategies they have implemented, providing a more relatable view. The local intranet can be used for user friendly access, navigation and collation of emerging information [39].

Using a piecemeal approach was another solution raised. The plethora of information on PA, both practitioner and patient prudent can be overwhelming but changing the narrative of how this is viewed is a starting point [35,37]. Implementing a simple, concise and gradual approach to information dissemination can be more effective for longer term adherence. This has been mirrored in data in the general public’s attitude towards starting PA, and thus shows a suitability [40]. Applying a “progression not perfection” strategy links with the subtheme of motivation, which can build confidence and offset inactivity effects, thus improving longer term PA levels. PA advice may be totally new to many patients and giving small but mountable tasks will help with confidence, compliance and greater opportunities for practitioners to build on their targets. Nurses themselves will benefit from this enhanced confidence by building their own knowledge base and learning practical tips for application when time and other clinical duties are in conflict. Using a non-judgmental approach is vital to maintain trust between staff, as many could blame themselves and become isolated. Overall, there are a wide array of simple and practical solutions raised by this study, but most importantly, recognising and educating the NHS nursing workforce remains the greatest-yield strategy. Even a small amount of information into offering online tools to support consultations such as CMO physical activity guidance, the Moving Medicine website, which provides a wealth of tailored information for exercise and illness, and wider advertising campaigns can largely impact practice. Suggestions included social media campaigns and radio and television adverts to make the topic itself more recognised. The popularity and viewership of these techniques will help to encompass a variety of societal cohorts, through age, gender and socioeconomic slants on the message itself. These resources could be centralised into a tab for consults which could be automatically sent to patients, thus facilitating a more in-depth, non-judgmental conversation.

### 4.3. Study Strengths and Limitations

Limitations include the small sample size, taken from a specific region(s) of the UK. There were greater numbers from the North of England, and therefore we cannot fully assess the insights on the more populous regions of London and the south. Further research could target a broader geographical location to improve this. Additionally, there was a greater risk of participant bias as those more likely to have interest in PA were forthcoming for the study. This may mean the results are not representative of the nursing population as a whole. The research team attempted to mitigate this by circulating the study information amongst a wide variety of nurses from different clinical experiences and made it clear to the participants that no prior knowledge was required to enter the study. Limitations also included potential researcher bias despite the best efforts to interpret the information. Limitations are balanced by study strengths which include the involvement of nurses in the identification of barriers and facilitators from the perspective of nurses. Nurses have been identified as key conduits for PA promotion and generating approaches which enhance their preparedness is key. As such their involvement in identifying challenges and developing solutions is key. Further, this study provided rich and informative learning opportunities that can be used to help shape interventions which can support nurses in PA advice delivery.

### 4.4. Future Research Directions

Future research could use these data to help develop a survey investigating this topic further across a wider geographical area. Further research could assess student nurses’ knowledge of PA guidance. Through highlighting these gaps early on, support can be given via undergraduate teaching to enhance the curriculum sufficiently. This remains a potential source for further issues in the future if not addressed. Regarding the impact of this study, findings from this research will be disseminated through a range of channels in order for colleagues to benefit from the learning gained through both the research process and the research outcomes. These channels and networks include the Regional Public Health Networks, the Regional Physical Activity Delivery Networks such as the Active Partners Trust/Making our Move, Move More Derby in the East Midlands and the UK CMO Physical Activity Communication Network, as well as the Moving Medicine Network.

## 5. Conclusions

This study provides a better understanding of nurses’ views on barriers and solutions in discussing PA with patients. The chief barriers identified were an insufficient knowledge of the PA guidelines and insufficient recognition of the professional role of nurses in applying these. Prioritising both PA education in undergraduate and postgraduate training and within daily clinical duties were identified as potential solutions. Integrating a formal PA module into routine undergraduate nursing programmes in the UK is not easy but is certainly a worthwhile endeavour. Applying standards such as formal, routine postgraduate teaching sessions, CPD points and Quality and Outcomes Frameworks would further cement this. Improved HCP education, awareness of the role of PA as a health intervention and access to appropriate resources, can help facilitate the necessary systematic change.

## Figures and Tables

**Table 1 ijerph-20-07113-t001:** Participant characteristics (*N* = 13).

Characteristic	Category	*N* (%)
Work experience (years)		
	1–5	2 (15.4)
	6–10	7 (53.8)
	11+	4 (30.8)
Job role		
	Staff Nurse	7 (53.8)
	Clinical Research Nurse	2 (15.4)
	Health Visitor	1 (7.7)
	Practice Nurse	2 (15.4)
	Senior Public Health Nurse	1 (7.7)
Field		
	Cardiology	1 (7.7)
	Cardiothoracic intensive care	1 (7.7)
	Gastroenterology	1 (7.7)
	Oncology	1 (7.7)
	COVID assessment wards	1 (7.7)
	Paediatrics	2 (15.4)
	Public Health	1 (7.7)
	Research Trials	2 (15.4)
	Undisclosed	3 (23.1)
UK Region		
	West Midlands	3 (23.1)
	Yorkshire and the Humber	2 (15.0)
	Southwest	1 (7.7)
	Scotland	3 (23.1)
	Undisclosed	4 (30.7)
Aware of CMO’s PA guidelines		
	Yes	3 (23.1)
	No	10 (76.9)
Aware of the 2019 PA guideline update		
	Yes	1 (7.7)
	No	12 (92.3)
Meeting CMO’s PA guidelines for aerobic activity		
	Yes	12 (92.3)
	No	1 (7.7)
Aware of Moving Medicine		
	Yes	2 (15.4)
	No	11 (84.6)

CMO; Chief Medical officer; *N*, number; PA, physical activity; UK, United Kingdom.

**Table 2 ijerph-20-07113-t002:** Themes and subthemes.

Theme	Subtheme
1—Nurses’ intrinsic barriers to delivering PA advice	Lack of knowledge of PA guidelines
	PA is often an afterthought
	Compartmentalisation
	Nurse’s guilt
2—Nurses’ extrinsic barriers to delivering PA advice	Lack of formal PA education
	Time pressures
	Patient engagement
3—Increasing staff awareness of PA guidelines	Staff training
	Incorporating PA learning via policy
	Encouraging staff to be active
4—Optimising PA advice delivery	Individualised PA advice
	Piecemeal Approach
	Interdisciplinary team approach
	Local exercise services and schemes
	Utilising online and visual resources

PA; physical activity.

**Table 3 ijerph-20-07113-t003:** Theme 1: Nurses’ intrinsic barriers to delivering PA advice.

Sub-Theme	Example Quotes
Lack of knowledge of PA guidelines	“I’d just be encouraging gentle exercise like walking and nothing that could potentially cause fractures or injury, but because my knowledge isn’t great in terms of exercise and what would be safe and what wouldn’t be, I’d probably be more cautious and tell people to avoid things that could cause them injury.” (Participant 4) “What I find quite difficult is that with every subject you have a little bit of knowledge and then you can go and find out more yourself. Yeah, and things change all the time as well. It’s not like, you know, there’s one initiative in. The next time you try implement it, it’s like actually that’s changed now.” (Participant 13)
PA is often an afterthought	“We may well have covered it, but it isn’t something that I’ve retained.” (Participant 10)
Compartmentalisation	“Confidence, I would say. I also think that we’ve become quite compartmentalised in our roles.” (Participant 11) “I’ve had this conversation with colleagues before. It seems like as nurses, we now feel, or can feel that is the physios’ job to give the exercise advice, and it’s the dietitian’s job to give the nutrition advice.” (Participant 5)
Nurses’ guilt	“There’s nothing worse than nurses’ guilt when they’ve just sat there and they’ve been like, ‘Oh I’ve done that’, or, ’That’s actually been detrimental to my patient if I made it push them harder’.” (Participant 3)

PA; physical activity.

**Table 4 ijerph-20-07113-t004:** Theme 2: Other barriers to delivering PA advice.

Subtheme	Example Quotes
Lack of formal PA education	“I don’t think I’ve really received an awful lot in terms of a lecture or an actual discussion.” (Participant 3) “I don’t remember anything from university really, in terms of education and educating patients. Yes, I guess it needs to be more ingrained in everything rather than one module that you do as a whole.” (Participant 4)
Time constraints	“I know a lot of my colleagues are a bit averse to bringing in more things and they think, oh, it’s another thing that we need to talk about. It’s another health promotion topic. I guess that can be a barrier, but I think time constraints is the biggest barrier and just trying to fit it all in.” (Participant 5) “How do they get stuff done? It’s very difficult to get them doing even day training or two-hour training.” (Participant 12)
Insufficient CPD training	“Even in my current role, we’ve never had any training at all in delivering advice for physical activity and stuff.” (Participant 1) “I don’t feel it’s something that we have a lot of training on. Although it’s something that everyone knows we should do, it doesn’t always mean people are doing it. Yes, I think more education is needed. Even in our nursing training I don’t feel we had lots of it We need more education for nursing staff to then make sure they’re confident enough to pass it on to the patients.” (Participant 2)
Patient engagement	“Some patients don’t take their illnesses very seriously.” (Participant 1) “We have a lot of patients who are younger and able to mobilize well, so we do make a point of encouraging them to go for walks around the wards or go for a walk around the hospital, or encourage them to do that quickly rather than—especially because they’re young and fit physically.” (Participant 2)

CPD; continuing professional development, PA; physical activity.

**Table 5 ijerph-20-07113-t005:** Theme 3: Increasing staff awareness of PA guidelines.

Subtheme	Example Quotes
Staff training	“We don’t really get enough training. I think maybe it’s because it’s so different, person to person, they don’t really know where to start. I don’t know if that’s wrong for me to say or not, but there’s maybe not specific guidance because you can’t specify physical activity person to person because it’s so varied.” (Participant 3) “Healthcare professionals going out training into a clinical area is the only way you can get staff trained. Getting staff off wards for any sort of training is a struggle and you really struggle to get people off wards.” (Participant 12)
Incorporating PA learning via policy	“Maybe even having a small program, a small e-learning module that takes 10 min that is—I know it’s totally out there but like included in your mandatory learning so that people will have to do it because it’s in your mandatories.” (Participant 1) “I suppose if they bring in policies, then if people get more information about it and if there are care plans or something that makes us aware of that, then, like I say, either part of the admission pack or part of the discharge planning.” (Participant 2)
Encouraging staff to be active	“I think in things like the Cycle to Work scheme and I know that some of the NHS discounts on gyms and leisure centres and things, there’s more accessibility and encouragement for staff.” (Participant 5) “As healthcare providers, what are we actually doing to encourage our own staff to be active and to keep healthy? If they see us as a big employer, as a trust actually supporting staff to be healthy and sort of making sure that they know it’s good for them to actually be physically active not only for their own health physically, but also for their mental health. I think it’s also as a trust, being mindful of the fact that we should be sort of encouraging our own staff to think about their own.” (Participant 12)

PA; physical activity.

**Table 6 ijerph-20-07113-t006:** Theme 4: Optimising PA advice delivery.

Subtheme	Example Quotes
Individualised PA advice	“He was running ultra-marathons before he had an MI. He was out doing a training run when he arrested. I was talking to him because I’m a runner as well. For him, he was talking about whether or not he’d be able to do the West Highland Way race again, and I was like, “Just take it gently, build up”, and so for him I adapted, I was able to have a full conversation with him about exercise.” (Participant 1) “Sometimes I feel if you have a discussion about it as well as maybe giving them a leaflet with the exercises system, I feel that would be more beneficial to have the conversation to go with it.” (Participant 3)
Piecemeal approach	“If people realised how little it is—even just being like, we’ll say we’ll do the minimum. We’ll do the 150 min, if you do that for a week and then they realise I’ve done over that or I’m so close to doing that anyway, then that might encourage people to actually do it. I think a lot of people don’t know that’s the guidelines. I certainly didn’t until I got my new watch. I think the more people you make aware of, so advertising it on social media that this is the guidelines. I think just doing it randomly doesn’t really help because I think if you make a big deal about it.” (Participant 1) “I just think of somebody who’s quite inactive, whereas if you maybe broke it down into a day by day, ‘This is what options you could do’, I think you’re probably going to have a better outcome than just giving them, ‘This is the amount of minutes that we want you to do’.” (Participant 3)
Interdisciplinary team approach	“As a nurse it would be important to be involved by making more of an interdisciplinary team approach; a discussion with a doctor and what they think would be suitable as well as the physios.” (Participant 3) “I guess more support from other health professionals might be helpful, more education. There’s definitely not enough focus on tackling the root of the issue.” (Participant 4)
Local exercise services and schemes	“I think asking what they do already or what they’d be interested in doing. I think trying to ascertain what their resources are available for exercise. If financially they don’t have money or access to gyms or swimming pools or wherever, encouraging things like walking or running if they’re interested, and then if they do have money to spend on things, then what would they be able to commit to?” (Participant 5)
Utilising online and visual resources	“Maybe having information. If patients are keen to find out that sort of stuff, having information leaflets that they can take home themselves. Promotion on television and radio and stuff is also always beneficial.” (Participant 2) “A lot of the trusts have their own Facebook pages. Like Facebook, they have so many followers, they basically have their whole—it’s not even just NHS workers that are in that. If you put on there the information, I think it would get seen by lots of people and with pictures” (Participant 7)

MI; myocardial infarction; NHS; National Health Services; PA; physical activity.

## Data Availability

The data presented in this study are available on request from the corresponding author. The data are not publicly available to avoid potential identification of interviewees based on comments made.

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
