# Peer review of "UK Nurses Delivering Physical Activity Advice: What Are the Challenges and Possible Solutions? A Qualitative Study"

_ijerph, 2023, doi:10.3390/ijerph20237113_

Round 1

Reviewer 1 Report

Comments and Suggestions for Authors

The topic of the paper is very interesting, because the impact of physical activity (PA) is important on the cardiovascular, movement and psychological systems.

Introduction

This study seeks to establish an understanding of nurse’s awareness of physical activity guidelines and their views on barriers and solutions.

Methods

The methodology uses only qualitative methods.

A qualitative study was conducted using semi-structured interviews.

The inclusion criteria for this study included nurses having an up to date valid licensing from the Nursing and Midwifery Council (NMC) and to be currently working in practice at the time of interview in any clinical field.

The interview ncluded questions regarding physical activity knowledge and education; physical activity. Participants were also asked about their years of work experience, region and working field.

Transcripts were chronologically analysed for key features for challenges and solutions as per the aim of the study. New information was analysed for via a recursive method.

The methodology for the qualitative study was followed.

Results

Over half of the nurses were confident that they could deliver the physical activity guidelines to patients. However, 77% of respondents professed that they were not aware of the physical activity guidelines.

The lack of knowledge of the physical activity guidelines was frequently cited, many expressed fear that they would give the incorrect advice.

Multiple barriers focusing on the lack of focus given to physical activity were raised by the nurses and 54% of respondents had received some form of physical activity -related education.

Discussion:

The nurses pointed to several barriers aimed at the lack of knowledge about physical activity, for example: lack of formal of physical activity education, staff training and  time constraints.

Discussion contain of barriers and solutions.

The bariers: Inadequate knowledge toward PA guidance was detected amongst the nursing cohort.

The solutions included formal staff training, staff support and encouragement, interdisciplinary working and other activities.

Examples include e-learning modules, formal, exercise proformas and others.

I recommend adding the latest studies on educational plans and physical activity programs for patients. I also recommend supplementing the possibilities and barriers of multidisciplinary cooperation of nurses and physiotherapists in the area of movement activities for patients.

Conclusions

It would be appropriate to indicate ways to improve nurses' knowledge of physical activity in undergraduate, postgraduate education and in clinical practice

I suggest formal and citation adjustments:

References

Association, B.M. Health funding data analysis. Available online: https://www.bma.org.uk/advice-and-support/nhs-556 delivery-and-workforce/funding/health-funding-data-analysis (accessed on 15/09/2023).

Digital, N.E. Quality and Outcomes Framework, 2020-21. Available online: https://digital.nhs.uk/data-and-605 information/publications/statistical/quality-and-outcomes-framework-achievement-prevalence-and-exceptions-data/2020-606 21#highlights (accessed on 15/09/2023).

Live, P. NHS intranets are a 'valuable clinical and professional resource'. Available online: 624 https://www.pmlive.com/digital_handbook/pharma_and_digital/digital_marketing/nhs_intranets_are_a_valuable_clinical625 _and_professional_resource (accessed on 15/09/2023).

Comments on the Quality of English Language

I recommend only formal edits (font style edits)

Author Response

Discussion: I recommend adding the latest studies on educational plans and physical activity programs for patients.  

Thank you for your suggestions. Regarding the information for patients, in the discussion we have included Making our Move which are established under the banner of the Uniting the Movement programme as well as other national PA networks. Moving Medicine is also mentioned which provides educational resources for HCP to use with patients 

Discussion: I also recommend supplementing the possibilities and barriers of multidisciplinary cooperation of nurses and physiotherapists in the area of movement activities for patients. 

Thank you for this comment. Unfortunately, is not entirely clear what is being requested here hence this has not been actioned. In addition, the focus of the paper is on PA being given by nurses working independently, as opposed to in MDT settings. This is due to most nurses working independently. Undoubtedly, nurses working independently to promote PA as well as other MDT members such as physiotherapists, midwives and doctors will hopefully increase PA levels overall increase PA. However, given the word count in our opinion inclusion of this is unlikely to add to the manuscript. 

Conclusions: It would be appropriate to indicate ways to improve nurses' knowledge of physical activity in undergraduate, postgraduate education and in clinical practice 

We have added the following to the conclusions: 

“Integrating a formal PA module into routine undergraduate nursing programmes in the UK is not easy but certainly a worthwhile endeavour. Applying standards such as formal routine postgraduate teaching sessions, CPD points and Quality and outcomes frameworks would further cement this. “ 

References: I suggest formal and citation adjustments (of 3 refs) 

Thank you for spotting that. We have amended those references. Refs 8, 9 and 31 in the manuscript. 

Reviewer 2 Report

Comments and Suggestions for Authors

The paper is well written. I don't think it is necessary to present the appendix attached at the end.

1. Add more content to the introduction so that it contains the main concept in one or two paragraphs. Please provide sufficient information in the introduction as to why a qualitative research approach to this topic is necessary.

2. Explain the contents of the interview guide in the material section.

3. Please describe the methodology in detail so that the characteristics of qualitative research can be clearly displayed in the materials and methods section.

4. Match the alignment of the table.

5. Can you draw the main concepts revealed in the results in one figure? It will enhance readers’ understanding of qualitative research.

Author Response

Appendices: The paper is well written. I don't think it is necessary to present the appendix attached at the end. The paper is well written.  

Thank you for your feedback, it is much appreciated. We have decided to keep the appendix as it was not raised by the other reviewers, and we feel it does add value to the manuscript for readers. 

Introduction: Add more content to the introduction so that it contains the main concept in one or two paragraphs.  

Thank you for this suggestion, however the authors feel the introduction provides both context and the aim of the research in a logical and sequential manner. We do appreciate your point that a short, succinct introduction is preferable. However, we feel the introduction currently covers relevant points. Furthermore, we were asked to elaborate on aspects on the introduction by other reviewers.  

Introduction: Please provide sufficient information in the introduction as to why a qualitative research approach to this topic is necessary. 

Thank you for your feedback. We have added the following: 

“There is a need for research into the barriers and facilitators that nurses encounter when providing PA advice. Previous studies, which have explored this in a range of healthcare professions have done so using a qualitative approach [16,17]. This is due to the essential human factors and complexities surrounding PA uptake and compliance issues. Here, greater insight is gained through qualitative methods which permit a greater candour from interviewees and enhance depth for discussion points [18].” 

Methods: Explain the contents of the interview guide in the material section. 

Thank you for your comment. The following is reported in the methods, which covers the interview guide and refers the reader to the appendix for further details of questions asked. 

‘The interview guide (see appendix) was adapted from Vishnubala et al. (2022) [17] to make questions specific to nurses and included questions regarding PA knowledge and education; PA resources and interventions as well as questions regarding the participant’s own PA in relation to the CMO PA guidance. Participants were also asked about their years of work experience, region and working field.” 

Methods: Please describe the methodology in detail so that the characteristics of qualitative research can be clearly displayed in the materials and methods section. 

We believe the methodology is clearly given, including the semi-structured interviews, how the interview questions were created, how the interviews were carried out and the thematic analysis approach to analysis following Braun and Clarke’s 6 steps. Please can clarification be provided as to what further information is required.  

Results: ? Match the alignment of the table. 

Apologies, it is unclear what this is referring to. Please could you clarify which table you are referring to? As far as we can see all tables are correctly labelled and identified in the text.  

Results: Can you draw the main concepts revealed in the results in one figure? It will enhance readers’ understanding of qualitative research. 

Thank you for your comment. We believe table 2 in the results, which presents a tabulated overview of the key themes and their associated subthemes, allows the reader to understand how the subthemes were derived. Utilising similar tables tends to be a common method for displaying themes and sub-themes in qualitative research results. If the reviewer could provide an example (reference) of where this has been done using a figure, we could look into using a figure instead. 

Reviewer 3 Report

Comments and Suggestions for Authors

Important study topic with good detail as to how this fits into the goal of increasing PA access by the general public through nursing engagement.

Line 37: Consider changing to “facing public engaging in physical activity”

Line 61: Consider rewording: “regular and comprehensive PA advice distribution” Entire sentence is challenging to understand.

Line 86: Break into two sentences

Line 114: Were recordings reviewed for accuracy? These automated transcription options are not fully reliable

Theme 1. Subtheme “lack of knowledge of PA guidelines and lack of formal PA education could be combined”

In order to support a full subtheme I would suggest at least 2 participant quotes (minimum). If not I would say there isn’t enough to support the subtheme. It is unusual to have this many subthemes.

Theme 2. Patient engagement – I think this may be beyond the scope of this paper. It is well known that PA behavior change is challenging with low uptake. This may be beyond the role  of the nurse in discussing PA knowledge and implementation.

Theme 3. I don’t believe these need to be subthemes. The main theme if the increase staff awareness and these are all suggested ways of doing so.

Theme 4. Similar statement as above for theme 4.

In order to support a full subtheme I would suggest at least 2 participant quotes (minimum). If not I would say there isn’t enough to support the subtheme. It is unusual to have this many subthemes.

I would expand on line 370. Volunteer bias, highly active nursing population, fields they are in and the importance of PA in those patient populations would affect their knowledge. I think this is highly telling as to what was learned from these individuals and whether they insights would work in the broader nursing population. In addition I think it would be worth discussing the areas the participants came from as there is likely greater PA knowledge and discussion among a nurse in cardiology for instance that COVID assessment.

Really great discussion. Broad exploration of the field and how novel findings tie into it.

Author Response

Line 37:  

Introduction: Consider changing to “facing public engaging in physical activity” 

Thank you for your comment. We have changed to "Despite this, there remain multiple challenges facing the public engaging in PA."  

Line 61:  

Introduction: Consider rewording: “regular and comprehensive PA advice distribution” Entire sentence is challenging to understand. 

Thank you for your comment, we have changed to "…there remain many barriers to the distribution of routine and comprehensive PA advice…". 

Line 86:  

Line 86: Break into two sentences 

Thank you for your suggestion. We have changed to: 

"Those who expressed interest emailed the interviewers directly and were subsequently sent further details including an information sheet. This outlined the aims of the study and what was expected of them, including the right to withdraw at any time. A consent form was also attached, which was signed by the participant and returned, then an interview date was then arranged". 

Results: Theme 1. Subtheme “lack of knowledge of PA guidelines and lack of formal PA education could be combined” 

Thank you for the suggestion. Our intention was to show that a lack of knowledge about the guidelines was more current e.g. present day due to lack of NHS standards and the latter (lack of education) showed a lack of formal PA education, which supplies a wider discussion point on education and formal teaching. This ultimately was the finding from interpreting the data.  

Results: Theme 2. Patient engagement – I think this may be beyond the scope of this paper. It is well known that PA behaviour change is challenging with low uptake. This may be beyond the role of the nurse in discussing PA knowledge and implementation. 

Thank you for your thoughts. This was a theme that regularly appeared in interviews, where nurses reported their experiences on why patients avoid PA and identified this as a barrier to delivering PA advice. We therefore think it is relevant to include as subtheme for extrinsic barriers. Ultimately, it was a subtheme generated from the data and therefore we believe appropriate to include. 

Results: Theme 3. I don’t believe these need to be subthemes. The main theme if the increase staff awareness and these are all suggested ways of doing so. 

Thank you for your suggestions. Theme 3 addresses education in PA and theme 4 addresses the delivery of PA in practice. We feel keeping these themes separate whilst outlining clear actionable subthemes will be useful to inform stakeholders. Ultimately in qualitative data the themes and subthemes are generated from the data, as was the case here and why the topics developed into separate themes.  

Results: Theme 4. Similar statement as above for theme 4. 

Thank you for your suggestions. Theme 3 addresses education in PA and theme 4 addresses the delivery of PA in practice. We feel keeping these themes separate whilst outlining clear actionable subthemes will be useful to inform stakeholders 

Results: In order to support a full subtheme, I would suggest at least 2 participant quotes (minimum). If not I would say there isn’t enough to support the subtheme. It is unusual to have this many subthemes. 

Thank you for this suggestion. We agree it would strengthen our justifications for the data analysis and formation of the themes and subthemes. We have included 2 quotes for each subtheme.  

Lines: XX (to add after final edits) 

I would expand on line 370. Volunteer bias, highly active nursing population, fields they are in and the importance of PA in those patient populations would affect their knowledge. I think this is highly telling as to what was learned from these individuals and whether they insights would work in the broader nursing population. In addition I think it would be worth discussing the areas the participants came from as there is likely greater PA knowledge and discussion among a nurse in cardiology for instance that COVID assessment.   

Thank you for your comment. While we are aware of the word count, we have added the following sentence to the below: 

Additionally, there was a greater risk of participant bias as those more likely to have interest in PA were forthcoming for the study. “This may mean the results are not representative of the nursing population as a whole.”